# Study on High Performance Polymer-Modified Cement Grouts

**Costas A. Anagnostopoulos * and Melina Dimitriadi** 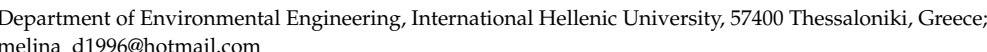

Department of Environmental Engineering, International Hellenic University, 57400 Thessaloniki, Greece; melina_d1996@hotmail.com
* Correspondence: kanagnos@cie.teithe.gr; Tel.: +30-2310-013872

**Abstract:** Engineers worldwide use various additives or chemical admixtures, such as polymer latexes, to improve the properties of cementitious materials for many construction projects. In this paper, the influence of acrylic or epoxy resin emulsions, along with a polycarboxylate superplasticiser on some basic properties (rheological behaviour, setting time, bleeding, strength) of thick cement grouts is presented. The experimental approach included the use of different polymer dosages mixed with grouts made of low water to cement ratios. The laboratory tests revealed that the incorporation of acrylic resin in grouts marginally affected the viscosity, whereas a significant increase in viscosity was obtained when an epoxy resin was added. Regardless of the prolonged setting times, both polymers improved the development of early or final strength. An acrylic resin dosage ranging from 0.25% to 0.75% and an epoxy resin dosage from 5% to 7.5% displayed the highest strength values, at all water to cement ratios. Additionally, all the polymer-modified grouts exhibited a higher bleed capacity, a fact that is significantly important where the bleeding of the grouts is crucial.

**Keywords:** polymer–cement blends; rheological properties; compressive strength

## 1. Introduction

The utilisation of a large number of organic admixtures in cement suspensions is vital for grouting technology. Admixtures contribute significantly to the production of grouts with superior physical or mechanical properties compared to those obtained with the use of neat cementitious materials. Their use leads to improved grouting effectiveness in many construction projects, that require soil improvement and foundation strengthening of seismically weak buildings, ground stabilisation for tunnel construction, grout injection in cracks for the restoration of concrete buildings and grouting of pre-stressed cables in pre-stressed concrete structures, as well as reinforcing the base plate of steel frame structures [1–5].

Nowadays, different chemical admixtures, for example, plasticisers, superplasticisers, viscosity-modifying agents, accelerators, anti-freezers, air-entrainers, volume stabilisers and numerous others, have been used extensively to adjust the properties of grouts as indicated by the worksite conditions and the aims of the grouting project [6–14]. Lately, there have been many types of water-soluble polymers and water-dispersed polymeric particles (latexes) being incorporated into mortars or concretes in order to improve their fluidity [15], mechanical properties [16], impact and abrasion resistance [17], waterproofness [18] and resistance to corrosive environments [19]. In particular, polymer–cement composites have demonstrated a discernible increase in mechanical strength because of the strong coherent matrix between the cement hydrate and the polymer membranes [20–22].

Acrylic resin (AR) is one of the common polymeric compounds used for the manufacturing of polymer-modified cementitious composites, such as mortars and concretes [23–28]. It is a colloidal solution of very small dispersed particles with sizes varying from 0.04 to 0.2 µm which are dispersed in water. AR particles are formed by an emulsion polymerisation process of a mixture of acrylic monomers. The monomers utilised are 2-hydroxyethyl methacrylate, methyl methacrylate (primary component), trimethylol propane triacrylate

and acrylic acid as cross-linking agents. Some studies have investigated the combined use of AR with superplasticisers for the further improvement of the strength properties of cement composite pastes [21,29].

Latex can likewise be made by utilising waterborne epoxy resin (ER) emulsions. ER emulsions are a suitable solution as a restoration material for old damaged concrete structures. Additionally, they are widely used with cement and aggregate because their incorporation in concrete or mortar has many advantages, such as improved workability and reduced segregation, increased mechanical properties and bond strength, increased ductility, reduced water penetration, reduced shrinkage, high resistance to freezing–thawing cycles and high resistance to acid environments [24,30–32].

Although much research has been carried out concerning polymer-modified concretes or mortars, there is little data available on the impact of latexes, either alone or combined with superplasticising admixtures, on some basic properties of cement grouts which are in direct relation to the efficacy of the grouting activity.

To build up on scientific knowledge related to the utilisation of polymer additives for the development of high-performance cement grouts, the fundamental goal of this experimental research was to explore the impact of AR or ER emulsions, along with a polycarboxylate high-range water reducer, on the bleeding, setting time, rheological behaviour and strength development of thick cement grouts produced utilising various water to cement (w/c) ratios.

## 2. Materials Used

The cement used in this laboratory investigation was a commercially available Portland cement classified as CEM I 52.5 N. It has a relative density of 3.1 and a specific surface area of 435 $m^2$/Kg.

A new generation polycarboxylate ether-type (PCE) dispersant was the chosen superplasticiser. Its main features are summarised in Table 1.

**Table 1.** The main features of the superplasticiser.

| | Polycarboxylate ether |
|---|---|
| Aspect | Slightly yellow |
| Specific gravity | 1.05 |
| pH | $6.3 \pm 0.5$ |
| Chloride ion content | Chloride free |
| Solid content | 40% |
| Molecular mass | 44,000 g/mol |
| Recommended dosage (% by cement mass) | 0.6–1.4 |

Epoxy resin (ER) is soluble in water and its base is the diglycidyl ether of bisphenol-A. As a curing agent for the resin, an aliphatic amine was utilised. It is resistant to water, alkalis, acids, petroleum products, etc. The ideal mixing ratio of epoxy resin (A) and hardener (B) is equal to A:B = 2.5 by weight. The manufacturer declares that the mixture of epoxy resin and hardener, without adding water, achieves its maximum strength after seven days of curing. The unconfined compressive, flexural and adhesive strength values are 70, 35 and 3 MPa, respectively.

As an acrylic resin polymer latex (AR), a commercial product of methyl methacrylate–acrylic acid copolymer was utilised. As an added substance in cement mixes, it significantly improves the bond between adhering concrete or mortars and the substratum surfaces, as well as the strength and cohesion of cement hydration products. It increases durability against chemical attack and freeze–thaw cycles. Additionally, it annihilates drying shrinkage and inhibits crack development. The polymer particle content of the acrylic emulsion is 35% by total weight.

### 3. Laboratory Procedure

Grouts were produced using ratios of w/c equal to 0.5, 0.4 and 0.33. The amount of superplasticiser (% by weight of cement) for the different w/c ratios corresponded to the dosage of saturation. For all grouts, the saturation dosages were estimated through measurements using the Marsh cone [33–35] and are shown in Table 2. The water content of the superplasticiser was accounted for in the total water content of the grouts. To study the influence of the ER (with or without hardener) or AR on the various properties of grouts, grouts were prepared with an ER to cement weight ratio from 0 to 10% and an AR to cement weight ratio from 0 to 1.5% (Table 2).

**Table 2.** Mix proportion design of the grouts.

| Designation | Proportion (w/c) | Superplasticiser (%) | Epoxy Resin + Hardener (%) | Epoxy Resin without Hardener (%) | Acrylic Resin (%) |
|---|---|---|---|---|---|
| $G_1$ | 0.5 | 0.5 | 0 | 0 | 0 |
| | | | 2.5 | - | - |
| | | | 5 | - | - |
| | | | 7.5 | - | - |
| | | | 10 | - | - |
| | | | - | 2.5 | - |
| | | | - | 5 | - |
| | | | - | 7.5 | - |
| | | | - | 10 | - |
| | | | - | - | 0.25 |
| | | | - | - | 0.5 |
| | | | - | - | 0.75 |
| | | | - | - | 1 |
| | | | - | - | 1.5 |
| $G_2$ | 0.4 | 1 | 0 | 0 | 0 |
| | | | 2.5 | - | - |
| | | | 5 | - | - |
| | | | 7.5 | - | - |
| | | | 10 | - | - |
| | | | - | 2.5 | - |
| | | | - | 5 | - |
| | | | - | 7.5 | - |
| | | | - | 10 | - |
| | | | - | - | 0.25 |
| | | | - | - | 0.5 |
| | | | - | - | 0.75 |
| | | | - | - | 1 |
| | | | - | - | 1.5 |
| $G_3$ | 0.33 | 1.5 | 0 | 0 | 0 |
| | | | 2.5 | - | - |
| | | | 5 | - | - |
| | | | 7.5 | - | - |
| | | | 10 | - | - |
| | | | - | 2.5 | - |
| | | | - | 5 | - |
| | | | - | 7.5 | - |
| | | | - | 10 | - |
| | | | - | - | 0.25 |
| | | | - | - | 0.5 |
| | | | - | - | 0.75 |
| | | | - | - | 1 |
| | | | - | - | 1.5 |

Epoxy resin cement grouts, with or without a hardener, were prepared with a blending sequence of two stages. The appropriate amounts of cement, water and superplasticiser

were added in the first stage to complete mixing for 2 min and in the second stage the desired quantity of ER, alone or in combination with the hardener (when combined with hardener the two components were stirred in a separate beaker), was added with an additional stirring for at least 3 min to attain a uniform mixture [36].

The production of the acrylic resin cement grouts was conducted by the simultaneous mixing of water, cement, superplasticiser and latex for at least 5 min to accomplish a uniform dispersion of cement particles. The one-stage mixing method was chosen instead of the two-stage mixing method in which the water and cement are blended first and latex is added afterwards in a later phase because of the fact that the delayed addition of latex in the cement mixture reduces its final strength, as indicated in past research [29]. As in the case of the superplasticiser, the water content of the acrylic resin solution was accounted for in the total water content of grouts to sustain a constant w/c ratio.

Preparation of the mixes was conducted by mechanical mixing with a high rotating stirrer recommended in ASTM C 938-10 and the setting time of the grout was estimated by performing Vicat needle tests as indicated by ASTM C953-10 at laboratory conditions of 23 $\pm$ 3 °C. In addition, sedimentation tests were performed in order to assess the bleeding of the grouts according to ASTM C940-10.

The development of mechanical strength was evaluated following unconfined compression and indirect (splitting) tensile strength tests on specimens aged for 3, 7, 30 and 90 days. Unconfined compression strength tests were conducted on cubic specimens with a side length of 50.8 mm. Specimens were subjected to compression loading at an axial strain rate of 0.1%/min. The elastic modulus was determined from the linear part of the stress–strain plot according to ASTM C469–10. Indirect strength tests were performed on cylindrical specimens with a diameter of 50 mm and a length of 100 mm according to ASTM D3967-05. The specimens were stored and cured as recommended by ASTM C109-12. Particularly, immediately upon completion of casting, test specimens were transferred into a moist cabinet with a temperature of 23 °C and relative humidity of 95%. After 24 h of curing, the specimens were de-moulded and immersed in saturated limewater in a storage tank until required for testing. The wet curing method was chosen in order to prevent the negative effect of drying shrinkage of specimens. To ensure reliable test results, the value of each of the reported mechanical properties is taken as the average value of at least three measurements, with a maximum allowable deviation of 5% from the average value.

The rheological behaviour and flow curves of the polymer-modified or unmodified grouts were assessed using a capillary tube viscometer to record the pressure–flow rate data [37,38]. The rheological measurements were performed at laboratory conditions of 23 $\pm$ 3 °C. A special facility was utilised for conducting the rheological experiments [39]. This facility was equipped with an air compressor, a mixing container (capacity of 50 L) with a stirrer rotating at high speed, flow and pressure meters, a pressure regulator, a transfer tube and a portion of a capillary plastic tube with a length of 1 m and an inner diameter of 5 mm (Figure 1). In order to avoid velocity profile distortion and to ensure consistent flow measurements, a flow meter was installed with a 10-tube diameter sufficient upstream and downstream straight-run piping which contained no fittings, valves or other obstructions. Additionally, the accuracy of the flow system was checked with the use of different fluids of known viscosity and was found to be satisfactory.

The flow curves were plotted with the shear strain rate ($\dot{\gamma}_w$) on the *x*-axis and the shear stress at the wall ($\tau_w$) on the *y*-axis. The $\tau_w$ and $\dot{\gamma}_w$ were calculated according to the Hagen–Poiseuille law for laminar flow as follows.

The suspension was permitted to flow through the tube with the application of constant pressure, and the resulting flow rate (Q) was recorded.

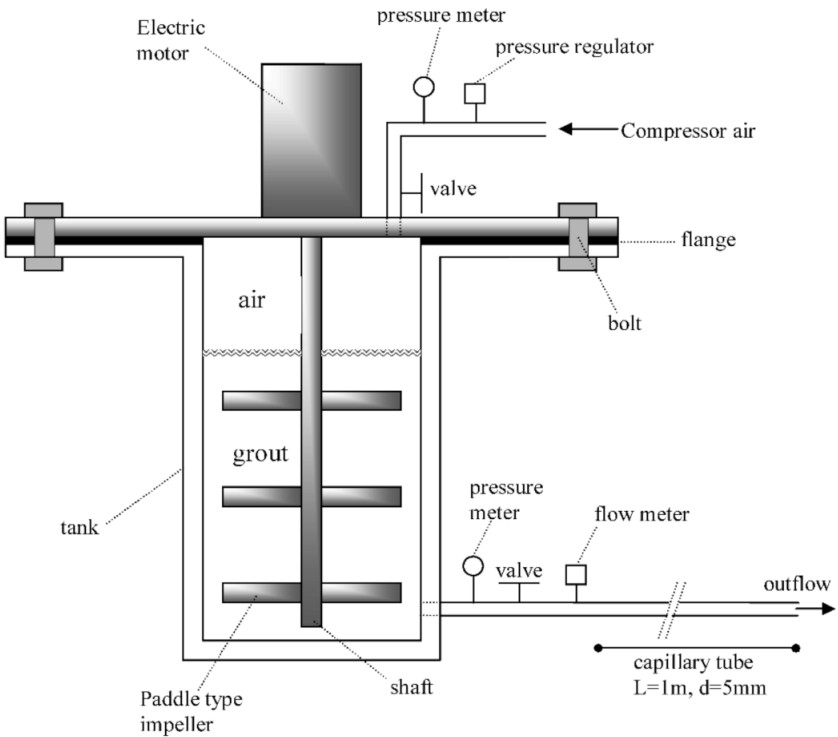

**Figure 1.** Apparatus for rheological experiments.

The mean velocity $V_{ave}$ of the grout is

$$V_{ave} = Q/S \tag{1}$$

The $\dot{\gamma}_w$ is calculated from the following equation

$$\dot{\gamma}_w = \left(\frac{dV}{dR}\right)_W = \frac{4 \cdot V_{ave}}{R} \tag{2}$$

Considering the balance of forces acting on a control volume of the fluid in the flow field, the expression for $\tau_w$ is obtained

$$\tau_w = \frac{\Delta P}{L} \cdot \frac{R}{2} \tag{3}$$

The apparent viscosity ($\mu$) is given

$$\mu = \tau/\dot{\gamma} \tag{4}$$

In Equations (1)–(3), S is the cross section of the capillary tube; $\Delta P$ is the pressure drop along the capillary tube; L is the length and R is the inner radius of the capillary tube.

To ensure reliability of the test results, each of the reported yield stress–shear rate values corresponds to the average value of at least three measurements that had values deviating no more than 1% from the average value of the total measurements.

## 4. Results and Discussion

Figures 2 and 3 depict the flow curves obtained from the rheological experiments on polymer-modified or unmodified superplasticised grouts. It is worth noting that the experimental results indicated a high shear thickening response (n >> 1) of $G_1$ grout without resin. An explanation for this observation is based on the formation of hydroclusters which significantly affect the rheological behaviour of grouts. The low concentration of superplasticiser, which induces electrostatic repulsion forces between cement particles, had as a result

the development of large compact groups of particles, especially at high velocity values. The higher the velocity (Peclet number), the higher the tendency for the particles to form clusters of large size [40], resulting in dissipation of a significant part of the applied energy because of the increased number of collisions between particles, and hence increased viscosity.

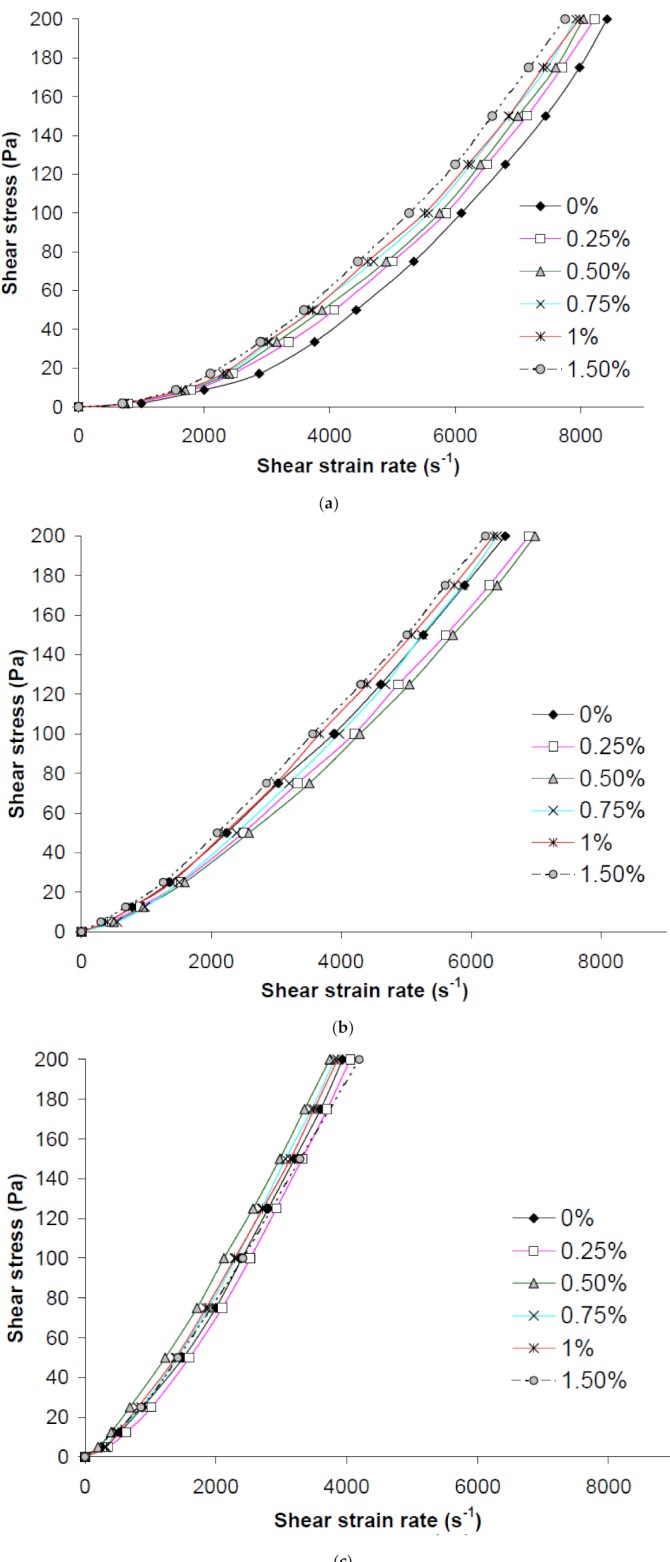

**Figure 2.** Flow curves of the unmodified grouts and acrylic resin-modified grouts (AMGs): (**a**) $G_1$; (**b**) $G_2$ and (**c**) $G_3$.

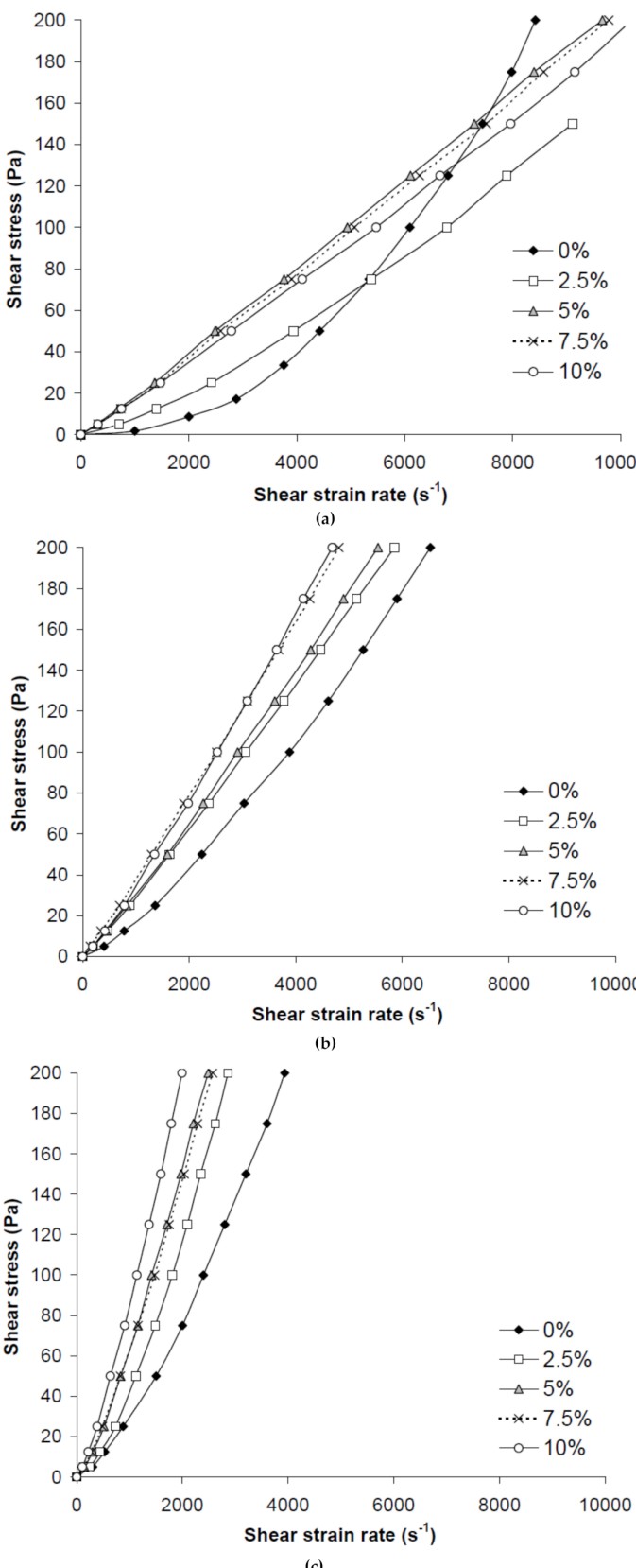

**Figure 3.** Flow curves of the unmodified grouts and epoxy resin-modified grouts (EMGs): (**a**) $G_1$; (**b**) $G_2$ and (**c**) $G_3$.

For all the grouts, our data revealed that their rheological behaviour (shear stress vs. shear strain rate relationship) was accurately described by the power law equation ($\tau =$

$k \cdot \dot{\gamma}^n$). The experimental results showed that the yield stress ($\tau_o$) of all of the grouts was very low or negligible and therefore it is not incorporated in the model. It should be noted that the presence of a hardener in the epoxy resin-modified grouts (EMGs) did not affect their rheological behaviour, which exhibited flow curves that were similar or identical to those of hardener-free grouts. The best-fitting curve for each cement grout composition, predicated on the previous model, was obtained by nonlinear regression analysis. In Tables 3 and 4, the parameters k and n are summarised, along with the correlation coefficients $R^2$.

**Table 3.** k, n parameters and $R^2$ for EMGs.

| Designation | Epoxy Resin Content (%) | k | n | $R^2$ |
|---|---|---|---|---|
| | 0 | $4 \times 10^{-7}$ | 2.21 | 0.99 |
| | 2.5 | 0.0008 | 1.33 | 0.99 |
| $G_1$ | 5 | 0.0125 | 1.06 | 0.99 |
| | 7.5 | 0.0102 | 1.08 | 0.99 |
| | 10 | 0.0113 | 1.06 | 0.99 |
| | 0 | 0.0019 | 1.32 | 0.99 |
| | 2.5 | 0.015 | 1.09 | 0.99 |
| $G_2$ | 5 | 0.016 | 1.1 | 0.99 |
| | 7.5 | 0.027 | 1.05 | 0.99 |
| | 10 | 0.012 | 1.15 | 0.99 |
| | 0 | 0.0017 | 1.41 | 0.99 |
| | 2.5 | 0.0014 | 1.49 | 0.99 |
| $G_3$ | 5 | 0.0071 | 1.31 | 0.99 |
| | 7.5 | 0.0113 | 1.25 | 0.99 |
| | 10 | 0.0151 | 1.25 | 0.99 |

**Table 4.** k, n parameters and $R^2$ for AMGs.

| Designation | Acrylic Resin Content (%) | k | n | $R^2$ |
|---|---|---|---|---|
| | 0 | $4 \times 10^{-7}$ | 2.21 | 0.99 |
| | 0.25 | $2 \times 10^{-6}$ | 2.03 | 0.99 |
| $G_1$ | 0.5 | $3 \times 10^{-6}$ | 1.99 | 0.99 |
| | 0.75 | $4 \times 10^{-6}$ | 1.99 | 0.99 |
| | 1 | $4 \times 10^{-6}$ | 1.98 | 0.99 |
| | 1.5 | $5 \times 10^{-6}$ | 1.96 | 0.99 |
| | 0 | 0.0019 | 1.32 | 0.99 |
| | 0.25 | 0.0013 | 1.35 | 0.99 |
| $G_2$ | 0.5 | 0.0009 | 1.39 | 0.99 |
| | 0.75 | 0.0004 | 1.49 | 0.99 |
| | 1 | 0.0016 | 1.34 | 0.99 |
| | 1.5 | 0.0042 | 1.23 | 0.99 |
| | 0 | 0.0017 | 1.41 | 0.99 |
| | 0.25 | 0.0007 | 1.51 | 0.99 |
| | 0.5 | 0.0068 | 1.25 | 0.99 |
| $G_3$ | 0.75 | 0.0018 | 1.41 | 0.99 |
| | 1 | 0.0016 | 1.43 | 0.99 |
| | 1.5 | 0.0036 | 1.31 | 0.99 |

The rheological behaviour of all the unmodified or acrylic resin-modified grouts (AMGs) was strongly shear thickening, and their apparent viscosity appeared to be enhanced with the increase in the applied pressure, as shown by the flow curves, irrespective of the AR content and the w/c ratio. This is consistent with the results reported in previous studies [35,41–43]. As indicated by the experimental results, the addition of AR to some extent modifies the rheological properties of all grouts, which appeared to be slightly more or less viscous in comparison to the unmodified grouts. In particular, $G_1$ and $G_3$

grouts with different AR content showed a mean increase in viscosity of 15.4% and 5.5%, respectively, whereas $G_2$ grouts showed an average reduction of 3%. These differentiations are attributed to the competition or synergy that takes place between PCE molecules and the chemical adsorption of AR particles on the grains of cement. The pore solution of fresh grout develops an alkaline environment, in which the carboxyl groups of the AR chain are hydrolysed and therefore negatively charged [26]. As stated in past research [44,45], the driving force for adsorption is the attraction between the positively charged surface of the mineral and the negatively charged surface of the polymer. When positively charged, as the cement hydrates are totally enwrapped by AR particles, they inversely obtain a negative load. This possibly explains why the viscosity of grouts is decreased by a low quantity of AR since the cement particles are repelled more strongly because of their negative charge, so they tend to disperse widely, leading to lower viscosity. When higher quantities of AR are used, the reason for the increase in viscosity could be the uneven adsorption of PCE molecules and polymer particles by the cement grains. Plank and Winter [46] demonstrated that when AR and PCE superplasticisers are gathered in the mortars, the AR particles and the PCE molecules competitively adsorb the cement grains. The AR particles possess higher zeta potential value, so they prevail in this competition [29]. Consequently, it is possible that cement hydrates adsorb a significant amount of the polymer particles and, as a result, a major quantity of non-adsorbed PCE is present in the free water which otherwise would be utilised for the dispersion of the cement conglomerates. Additionally, the unabsorbed or ineffectively adsorbed portion of the PCE polymer on the cement particles could initiate depletion flocculation [47] and, as a result, the osmotic balance could vary locally. However, the osmotic balance represents the main factor that supports the tendency of cement particles to flocculate or form compact clusters of particles. In this manner, these clusters capture extra water, resulting in higher friction between particles that augments the viscosity of the system [35]. Additionally, in the case of grouts with a very elevated concentration of cement particles, adjacent cement particles come closer together and the carboxylate units ($COO^-$) on the AR molecular chain tend to react with $Ca^{2+}$ ions. These two factors, when combined, may possibly drive the AR to cross-link with cement particles [48]. The concentration of the $Ca^{2+}$ particles around the cement grains increases. This impels interacting particles to assemble in enormous groups, which result in a larger increase in the system's viscosity.

In the case of EMGs, the rheological response of all w/c ratios was significantly affected by the addition of ER. Predominantly, the viscosity increased analogously to the amounts of ER, although some scatter was observed. In particular, $G_1$ grouts when ER was added displayed an increase in viscosity up to a certain value of strain rate beyond which the viscosity appeared to decrease to smaller values than those in the unmodified grout. $G_2$ and $G_3$ grouts demonstrated much higher viscosity values, compared with the unmodified grouts, for the whole range of strain rates. For example, the viscosity values of a $G_3$ grout without ER appeared to range between 0.016 and 0.051 Pa s when the shear stress was increased from 5 to 200 Pa; however, when a proportion of 10% ER was added to this grout, the viscosity increased between 0.05 and 0.1 Pa s.

The adsorption of ER molecules on cement particles could explain the apparent increase in the EMGs' viscosity compared to the unmodified grouts. The ammonium ions ($NH_4^+$) derived from the amines (hardener) or the hydroxide ions ($OH^-$) stemming from the alkaline environment of cement hydrates render the cyclic functional groups of oxacyclopropane, located at the brims of epoxy resin molecules, prone to becoming unlocked. As a result, two hydroxide groups are formed which distort the electrical charge leading to the formation of dipoles in the adjacent area. The ($OH^-$) groups of these polarised elements are responsible for their tendency to adsorb water on the surfaces of the positively charged cement particles. This adsorption creates a partial neutralisation effect, as reported by Olhero et al. [49], after the conducted zeta potential measurements. As a result, unadsorbed PCE polymer is probably present in large amounts in the pore solution otherwise used for the dispersion of the cement agglomerates. Moreover, as

in the case of AR, the unabsorbed or ineffectively adsorbed portion of the PCE polymer on the cement particles may initiate depletion flocculation [47] as a result of the variation in the nearby osmotic pressure that drives the cement particles to flocculate or form clusters. Thus, additional water, originally purposed to lubricate the system, is entrapped and increased inter-particle friction occurs, raising the viscosity of the grout [50,51].

Tables 5–7 depict the indirect tensile strength, compressive strength and elastic modulus of the different unmodified or AMGs with curing times increasing up to 90 days. The retarding action of AR on the setting and hardening of grouts appears to not adversely affect early strength development (3 and 7 days) in the majority of the grouts, which substantially exceeded that of the unmodified grouts. Notably, the final strength is not as significant in certain applications as early strength development, for example, in the grouting and filling of the tendons of cables in post-tensioned pre-stressed concrete members. This investigation's mechanical experiments demonstrated that AMGs could be utilised in such applications.

**Table 5.** Development of strength properties of the unmodified grouts and AMGs for water to cement (w/c) ratio = 0.5.

| | 0% | | | | 0.25% | | | | 0.5% | | | |
|---|---|---|---|---|---|---|---|---|---|---|---|---|
| | Curing Age (Days) | | | | Curing Age (Days) | | | | Curing Age (Days) | | | |
| Property | 3 | 7 | 30 | 90 | 3 | 7 | 30 | 90 | 3 | 7 | 30 | 90 |
| Compressive Strength (MPa) | 34.2 | 40.1 | 48.4 | 50.2 | 33 | 38.6 | 60.5 | 74.7 | 28.8 | 36.4 | 55.9 | 73.1 |
| Elastic Modulus (GPa) | 4.4 | 4.7 | 5.2 | 5.4 | 3.5 | 4.6 | 6.3 | 7.2 | 4 | 4.7 | 6.4 | 7.4 |
| Splitting Tensile Strength (MPa) | 2.6 | 2.8 | 3.7 | 3.9 | 3.1 | 3.4 | 4.5 | 5.7 | 3 | 3.2 | 4.5 | 5.6 |
| | 0.75% | | | | 1% | | | | 1.5% | | | |
| | Curing Age (Days) | | | | Curing Age (Days) | | | | Curing Age (Days) | | | |
| Property | 3 | 7 | 30 | 90 | 3 | 7 | 30 | 90 | 3 | 7 | 30 | 90 |
| Compressive Strength (MPa) | 28.3 | 37.6 | 51.5 | 70.8 | 35.5 | 41 | 50.4 | 73.2 | 33.7 | 45.8 | 52.2 | 74 |
| Elastic Modulus (GPa) | 3.9 | 4.8 | 6.5 | 7.6 | 5.1 | 5.6 | 6.6 | 7.6 | 4.57 | 6.3 | 6.8 | 7.7 |
| Splitting Tensile Strength (MPa) | 2.8 | 3.4 | 4.3 | 4.8 | 2.6 | 3.3 | 4.4 | 4.8 | 2.5 | 3.9 | 4.3 | 4.9 |

**Table 6.** Development of strength properties of the unmodified grouts and AMGs for w/c = 0.4.

| | 0% | | | | 0.25% | | | | 0.5% | | | |
|---|---|---|---|---|---|---|---|---|---|---|---|---|
| | **Curing Age (Days)** | | | | **Curing Age (Days)** | | | | **Curing Age (Days)** | | | |
| Property | 3 | 7 | 30 | 90 | 3 | 7 | 30 | 90 | 3 | 7 | 30 | 90 |
| Compressive Strength (MPa) | 55.2 | 60.2 | 69.9 | 71 | 60.5 | 66 | 85.2 | 116.5 | 59 | 76 | 90.6 | 123.3 |
| Elastic Modulus (GPa) | 6.2 | 6.5 | 7.2 | 7.3 | 7.1 | 7.3 | 8.4 | 10.4 | 6.9 | 7.8 | 8.6 | 11.4 |
| Splitting Tensile Strength (MPa) | 3.1 | 3.8 | 4.9 | 5.7 | 3.59 | 4.5 | 6.3 | 6.8 | 3.4 | 4.1 | 6.8 | 7.3 |
| | **0.75%** | | | | **1%** | | | | **1.5%** | | | |
| | **Curing Age (Days)** | | | | **Curing Age (Days)** | | | | **Curing Age (Days)** | | | |
| Property | 3 | 7 | 30 | 90 | 3 | 7 | 30 | 90 | 3 | 7 | 30 | 90 |
| Compressive Strength (MPa) | 59.1 | 69.9 | 93.2 | 125 | 68.1 | 72.3 | 98 | 126.9 | 64.9 | 78.9 | 104.7 | 123.5 |
| Elastic Modulus (GPa) | 6.8 | 7.4 | 8.8 | 11.7 | 7.4 | 7.6 | 9.1 | 11.9 | 7.5 | 9.1 | 10 | 12.1 |
| Splitting Tensile Strength (MPa) | 3 | 4.3 | 6.4 | 7.7 | 3.1 | 4.1 | 6.8 | 7.5 | 3.6 | 4.6 | 5.9 | 7 |

**Table 7.** Development of strength properties of the unmodified grouts and AMGs for w/c = 0.33.

| | 0% | | | | 0.25% | | | | 0.5% | | | |
|---|---|---|---|---|---|---|---|---|---|---|---|---|
| | **Curing Age (Days)** | | | | **Curing Age (Days)** | | | | **Curing Age (Days)** | | | |
| Property | 3 | 7 | 30 | 90 | 3 | 7 | 30 | 90 | 3 | 7 | 30 | 90 |
| Compressive Strength (MPa) | 62.9 | 73.6 | 91 | 94 | 70.9 | 80 | 95.1 | 130.3 | 70 | 76.4 | 102.7 | 132 |
| Elastic Modulus (GPa) | 6.5 | 7.2 | 9.4 | 9.5 | 7.5 | 8.7 | 9.9 | 11 | 7.9 | 8.8 | 10.7 | 11.6 |
| Splitting Tensile Strength (MPa) | 3.6 | 4.3 | 5.7 | 6.4 | 4.8 | 5.9 | 6.9 | 7.6 | 4.7 | 5.3 | 7.2 | 8.1 |
| | **0.75%** | | | | **1%** | | | | **1.5%** | | | |
| | **Curing Age (Days)** | | | | **Curing Age (Days)** | | | | **Curing Age (Days)** | | | |
| Property | 3 | 7 | 30 | 90 | 3 | 7 | 30 | 90 | 3 | 7 | 30 | 90 |
| Compressive Strength (MPa) | 68.3 | 78.8 | 104.2 | 131.4 | 62.9 | 76 | 106.7 | 127.5 | 64.5 | 72.7 | 109 | 115.7 |
| Elastic Modulus (GPa) | 8.2 | 9.2 | 10.5 | 12.1 | 8.2 | 8.9 | 10.5 | 12.2 | 8.2 | 8.93 | 11.1 | 12 |
| Splitting Tensile Strength (MPa) | 4.4 | 5.2 | 7.9 | 8.9 | 3.9 | 5.1 | 8.7 | 9 | 2.67 | 4.7 | 7.8 | 9.2 |

Our results showed that, over time, the strength of modified grouts tends to increase, which leads to substantially elevated strength values compared to those of unmodified grouts, particularly following 30 and 90 days of curing [43]. AR dosages ranging from 0.25 to 0.75%, at all w/c ratios, displayed the highest 30- and 90-day strength values. Beyond the 0.75% dosage, the increase in strength was negligible or even a reduction in strength was

observed. For instance, the addition of 0.5% AR to the $G_1$, $G_2$ and $G_3$ grouts resulted in a 90-day compressive strength increase of 45.6%, 73.6% and 40.4%, respectively, in comparison to the strengths of the unmodified grouts, while adding 1.5% AR caused an increase in 90-day strength of 47.4%, 73.9% and 23%, respectively, compared to the strengths of the unmodified grouts. This general image of the impact of AR on the mechanical properties of grouts and particularly the significant increase in strengths when small quantities of AR are added is in accordance with the test results over a wide range of cement types and w/c ratios of past investigations of polymer-modified cementitious materials that revealed similar findings [21,26,29,36,52].

The effects of the addition of ER on the compressive strength, indirect tensile strength and elastic modulus of the different grouts are shown in Tables 8–10.

**Table 8.** Development of strength properties of the unmodified grouts and EMGs for w/c = 0.5.

| | 0% | | | | 2.5% | | | | 5% | | | |
|---|---|---|---|---|---|---|---|---|---|---|---|---|
| | Curing Age (Days) | | | | Curing Age (Days) | | | | Curing Age (Days) | | | |
| Property | 3 | 7 | 30 | 90 | 3 | 7 | 30 | 90 | 3 | 7 | 30 | 90 |
| Compressive Strength (MPa) | 34.2 | 40.1 | 48.4 | 50.2 | 34.5 | 41.2 | 50.2 | 72 | 33.7 | 44.3 | 51.3 | 76.3 |
| Elastic Modulus (GPa) | 4.4 | 4.7 | 5.2 | 5.4 | 4.9 | 5 | 5.9 | 6.9 | 4.3 | 5.2 | 5.9 | 7.4 |
| Splitting Tensile Strength (MPa) | 2.6 | 2.8 | 3.7 | 3.9 | 2.7 | 2.9 | 3.8 | 5.4 | 2.8 | 3 | 3.9 | 5.9 |
| | 7.5% | | | | 10% | | | | | | | |
| | Curing Age (days) | | | | Curing Age (Days) | | | | | | | |
| Property | 3 | 7 | 30 | 90 | 3 | 7 | 30 | 90 | | | | |
| Compressive Strength (MPa) | 33.8 | 41.9 | 49.7 | 61 | 33.4 | 40 | 46.1 | 59.7 | | | | |
| Elastic Modulus (GPa) | 4.08 | 4.8 | 5.6 | 6.1 | 4 | 4.6 | 5.24 | 5.8 | | | | |
| Splitting Tensile Strength (MPa) | 2.7 | 2.9 | 3.8 | 4.7 | 2.6 | 2.8 | 3.6 | 4.5 | | | | |

**Table 9.** Development of strength properties of the unmodified grouts and EMGs for w/c = 0.4.

| | 0% | | | | 2.5% | | | | 5% | | | |
|---|---|---|---|---|---|---|---|---|---|---|---|---|
| | Curing Age (Days) | | | | Curing Age (Days) | | | | Curing Age (Days) | | | |
| Property | 3 | 7 | 30 | 90 | 3 | 7 | 30 | 90 | 3 | 7 | 30 | 90 |
| Compressive Strength (MPa) | 55.2 | 60.2 | 69.9 | 71 | 61.8 | 66.7 | 86.4 | 126.7 | 53.8 | 70.3 | 97.4 | 142 |
| Elastic Modulus (GPa) | 6.2 | 6.5 | 7.2 | 7.3 | 6.9 | 7.7 | 9.6 | 10.7 | 5.8 | 8.1 | 9.8 | 11.7 |
| Splitting Tensile Strength (MPa) | 3.1 | 3.8 | 4.9 | 5.7 | 3.4 | 4.3 | 5.9 | 7.8 | 3.4 | 4.5 | 6.6 | 10.2 |

| | 7.5% | | | | 10% | | | |
|---|---|---|---|---|---|---|---|---|
| | Curing Age (Days) | | | | Curing Age (Days) | | | |
| Property | 3 | 7 | 30 | 90 | 3 | 7 | 30 | 90 |
| Compressive Strength (MPa) | 58.9 | 64.4 | 90 | 132.2 | 53.5 | 60.4 | 87 | 126.5 |
| Elastic Modulus (GPa) | 6.4 | 7.7 | 9.3 | 11.4 | 6.1 | 7.5 | 9.2 | 11.2 |
| Splitting Tensile Strength (MPa) | 3.5 | 4.1 | 6.2 | 9.4 | 3.3 | 4.1 | 6.2 | 8.4 |

**Table 10.** Development of strength properties of the unmodified grouts and EMGs for w/c = 0.33.

| | 0% | | | | 2.5% | | | | 5% | | | |
|---|---|---|---|---|---|---|---|---|---|---|---|---|
| | Curing Age (Days) | | | | Curing Age (Days) | | | | Curing Age (Days) | | | |
| Property | 3 | 7 | 30 | 90 | 3 | 7 | 30 | 90 | 3 | 7 | 30 | 90 |
| Compressive Strength (MPa) | 62.9 | 73.6 | 91 | 94 | 66 | 76.6 | 91.7 | 114.6 | 72 | 78 | 93.4 | 122.3 |
| Elastic Modulus (GPa) | 6.5 | 7.2 | 9.4 | 9.5 | 7.4 | 8.4 | 9.6 | 10.8 | 7.9 | 8.9 | 9.8 | 11 |
| Splitting Tensile Strength (MPa) | 3.6 | 4.3 | 5.7 | 6.4 | 4 | 4.6 | 6.1 | 7 | 4.9 | 5.2 | 6.4 | 8 |

| | 7.5% | | | | 10% | | | |
|---|---|---|---|---|---|---|---|---|
| | Curing Age (Days) | | | | Curing Age (Days) | | | |
| Property | 3 | 7 | 30 | 90 | 3 | 7 | 30 | 90 |
| Compressive Strength (MPa) | 79.1 | 84.6 | 95.2 | 128.7 | 83.3 | 94.5 | 101.2 | 132.9 |
| Elastic Modulus (GPa) | 8.5 | 9.8 | 10.7 | 11.7 | 9.1 | 10.2 | 11 | 12 |
| Splitting Tensile Strength (MPa) | 5.4 | 5.8 | 6.8 | 8.7 | 6.1 | 6.5 | 7.3 | 9.5 |

As in the case of AR, despite the retarding action of ER, the EMGs exhibited greater early strength development than the unmodified grouts, except in a few cases. The strength tended to increase with the maturation of the EMGs, leading to substantially elevated 30- and 90-day strengths compared to those of the unmodified grouts. In general, the ER content, w/c ratio and curing time highly influence the increase in the strength values. The highest 30- and 90-day strength values were obtained for an ER dosage of 5% for grouts with w/c ratios of 0.5 and 0.4, beyond which a reduction in strength was observed. For

grouts with a w/c ratio of 0.33, the optimal dosage of ER is assumed to be 7.5%, since beyond this dosage the increase in strength was negligible. For instance, adding 5% ER to the $G_1$ grouts increased the 90-day compressive strength, the indirect tensile strength and the elastic modulus by 52%, 50.5% and 39%, respectively, whereas the same dosage of ER added to the $G_2$ grouts displayed corresponding increases of 100%, 80.5% and 60% for the same strength parameters when compared to the strengths of the unmodified grouts. However, the level of improvement in strength of the thicker $G_3$ grouts was limited. The optimal dosage of 7.5% increased the 90-day compressive strength, indirect tensile strength and elastic modulus by 37%, 35% and 22.6%, respectively. The aforementioned results can be explained by the formation of polymer films that caused a higher reduction in void space in the thinner grouts, which have a more porous structure than the thicker grouts and, as a result, a higher increase in strength of the thinner grouts [24,41,53,54]. It is apparent that the addition of ER refines the pore space in cement hydration products due to the filling effect of the formed polymer membranes. However, previous research conducted by Pourchez et al. [55] indicated a tendency for grouts of high viscosity, as in the case of the $G_3$ grouts, to cause high air entraining, introducing additional pores into the cement mixture and negatively influencing strength development. Additionally, another reason for the limited improvement in strength could be the high quantities of ER present in a thick mixture, which significantly restrict both the adsorption of PCE by cement particle surfaces and, consequently, the uniform dispersion of cement agglomerates [29].

The research and development of epoxy-modified mortars and concretes has advanced great deal and ER without any hardener in epoxy-modified cement mixtures has been reported to possibly be hardened when alkalis or hydroxide ions ($OH^-$) from calcium hydroxide [$Ca(OH)_2$] are present as one of the cement hydrates. The produced mortar or concrete is also superior in mechanical properties compared to unmodified ones [56,57]. One of the purposes of this experimental study was to assess the influence of hardener-free ER on the strength parameters of all tested grouts. The strength development of grouts was examined at 90 days of curing. The reason for this is that the hardener-free epoxy resin needs time to get a higher degree of polymerisation and hardening, as found in previous research [57–60]. Figures 4–6 show the compressive strength, indirect tensile strength and elastic modulus of the EMGs with and without a hardener after 90 days of curing. Indeed, the test results clearly show that the strengths of EMGs without the hardener are much higher than those of the unmodified grouts but less than that with the hardener. In the case of the $G_1$ and $G_3$ grouts, the strengths increased as the ER–cement ratio increased, whereas for the $G_2$ grouts, the highest strength values were attained with the 7.5% ER dosage, beyond which a slight reduction was observed. Therefore, it is postulated that the ER can harden in the presence of the alkalis in the cement grouts and significantly improve the strength of grouts. The work of Jo [57] demonstrated the significant influence of the addition of ER on the mechanical behaviour of mortars. Accordingly, after one year of curing, the flexural and compressive strength of epoxy-modified mortars without the hardener were 1.5 and 1.4 times higher than those with the hardener, respectively. It is most likely that a similar trend may be found in the case of cement grouts and further research is necessary to assess the strength parameters of epoxy-modified specimens cured for periods much longer than 90 days.

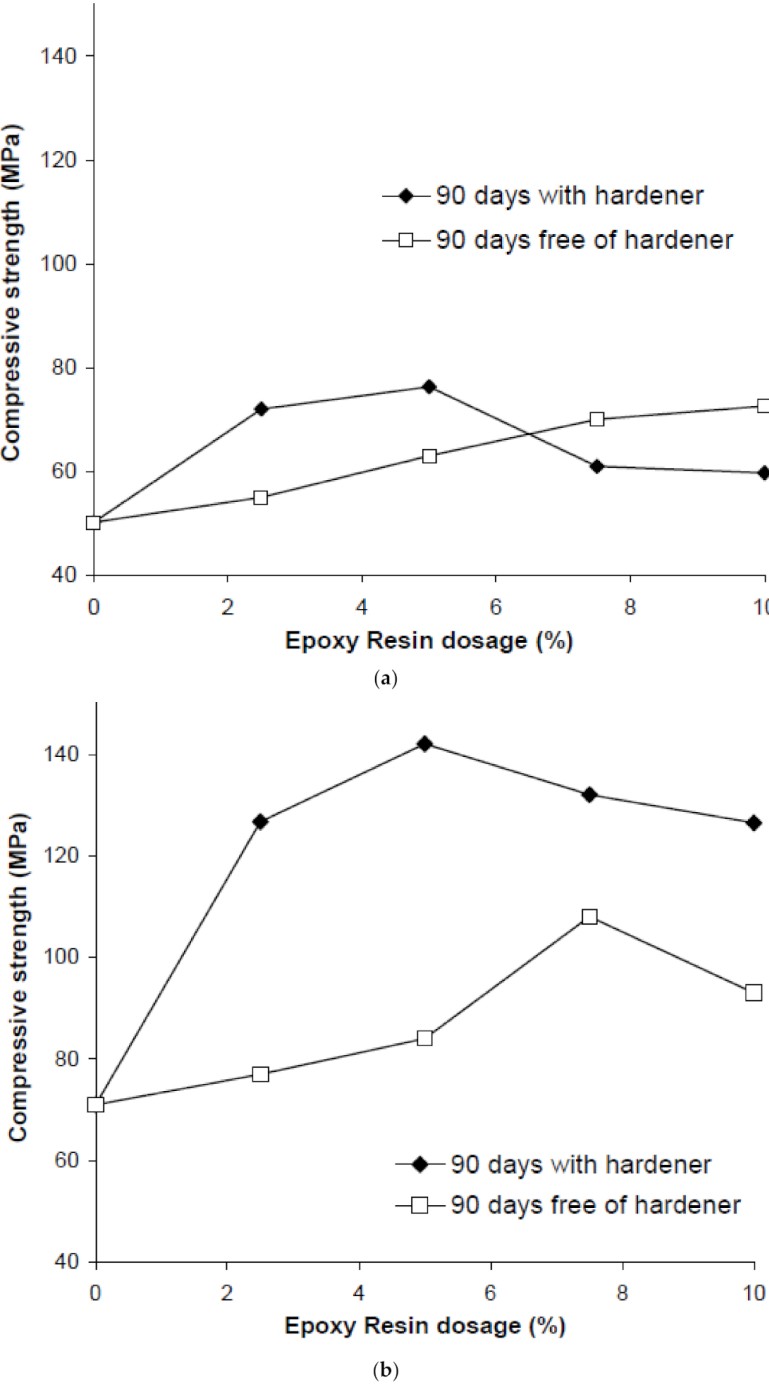

**Figure 4.** *Cont.*

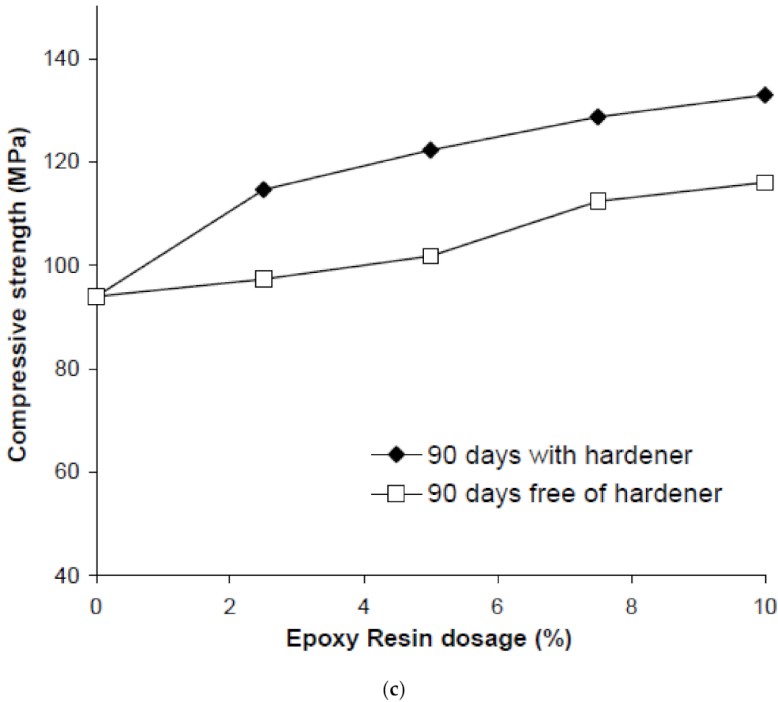

(c)

**Figure 4.** Compressive strength of EMGs, with and without a hardener at 90 days of curing: (**a**) G$_1$; (**b**) G$_2$ and (**c**) G$_3$.

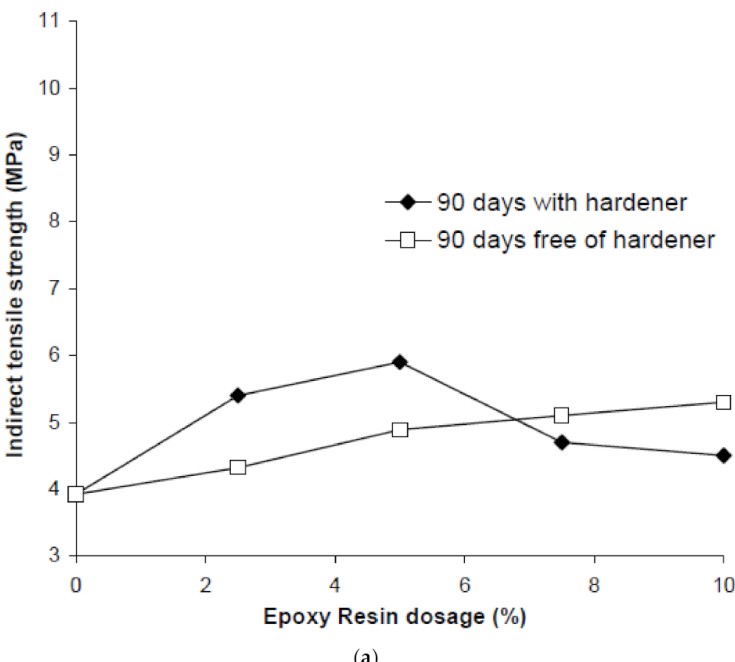

(a)

**Figure 5.** *Cont.*

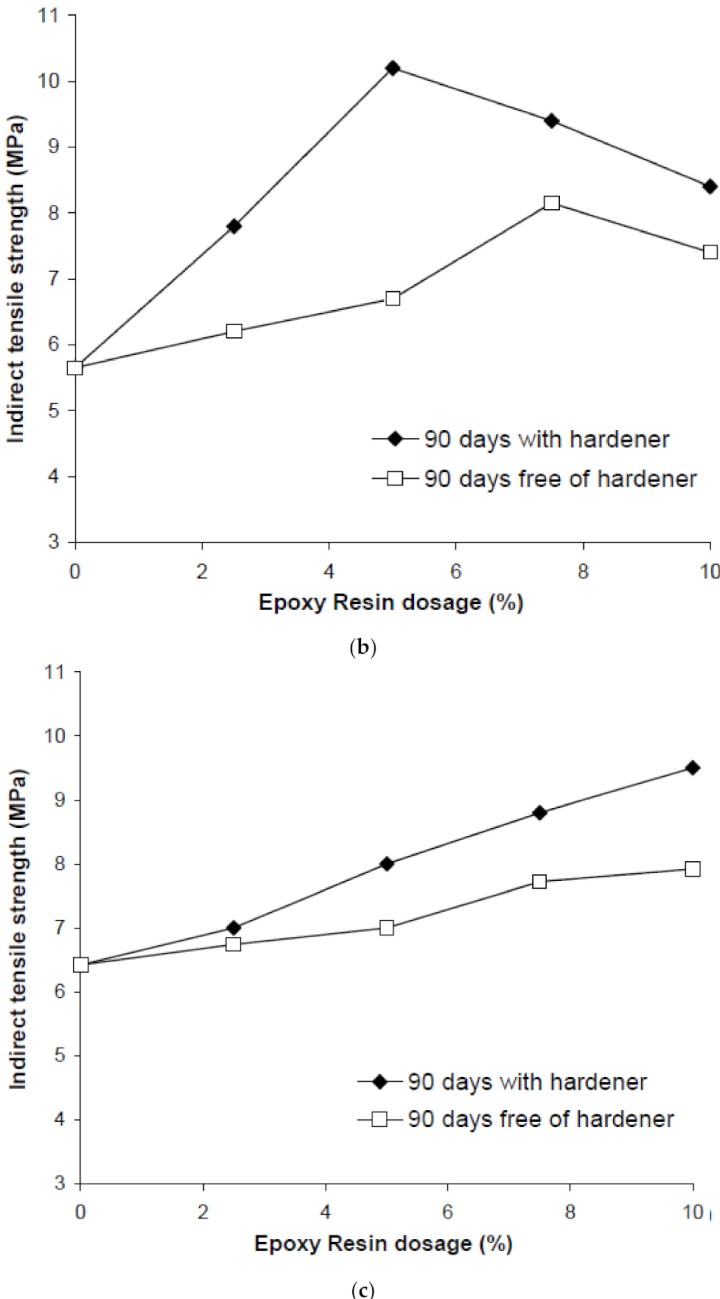

**Figure 5.** Indirect tensile strength of EMGs, with and without a hardener at 90 days of curing: (**a**) $G_1$; (**b**) $G_2$ and (**c**) $G_3$.

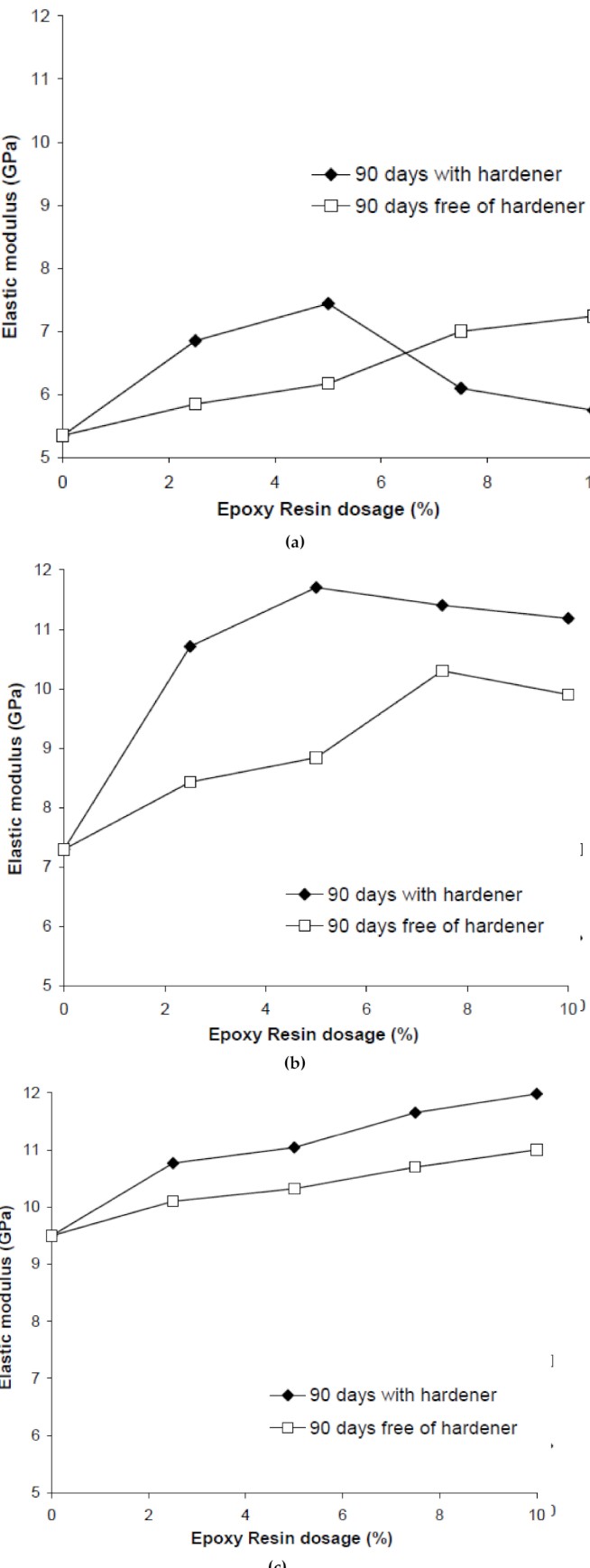

**Figure 6.** Elastic modulus of EMGs, with and without a hardener at 90 days of curing: (**a**) $G_1$; (**b**) $G_2$ and (**c**) $G_3$.

In total, the results of the test conducted in the current study showed a remarkable improvement of the elastic modulus, after the addition of AR or ER (with or without a hardener). This outcome is interesting in light of the fact that it is contrary to the outcomes obtained from past research [24,61], which postulate that polymers generally reduce the elastic modulus of cementitious material. In contrast, the results of the current study are in agreement with other studies [39,62] which demonstrate that polymer-modified cementitious materials tend to exhibit a higher elastic modulus than the unmodified material.

Figures 7 and 8 display the bleed capacity values of all the tested grouts.

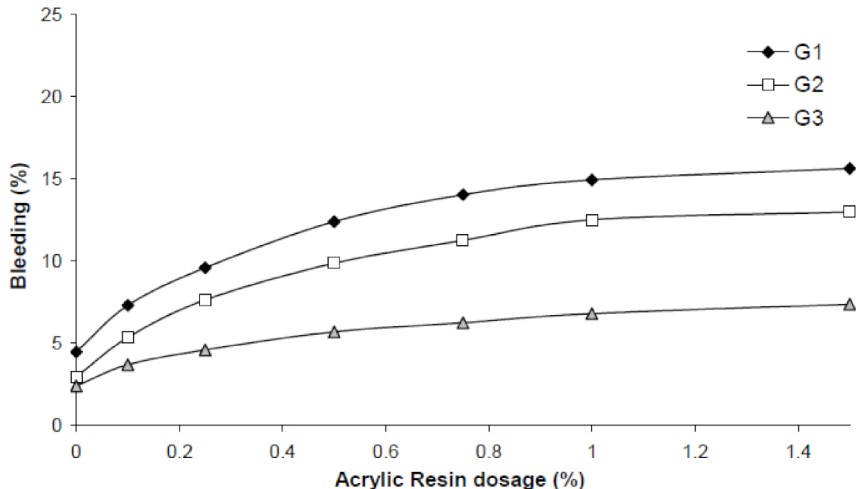

**Figure 7.** Bleeding of the unmodified grouts and AMGs.

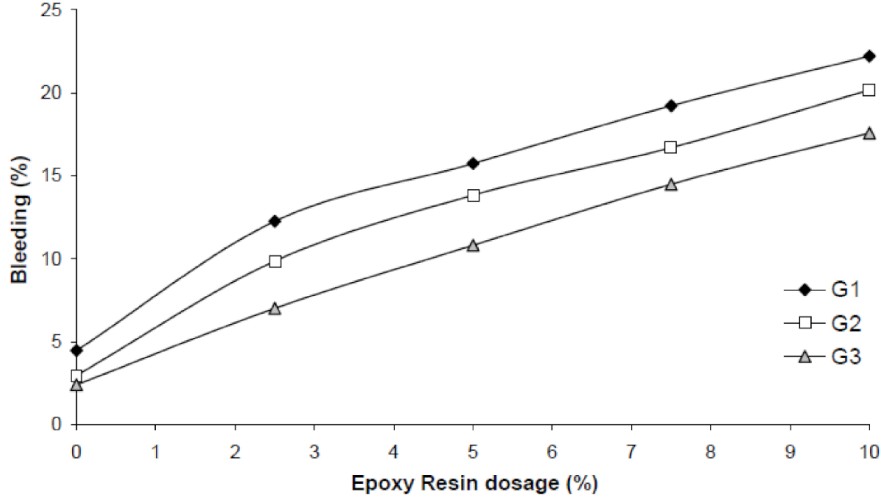

**Figure 8.** Bleeding of the unmodified grouts and EMGs.

It is easily noticeable that the addition of ER or AR substantially increased the bleeding of all of the grouts. The increase in polymer content resulted in an analogous increase in bleeding. Bleeding was more pronounced in the case of ER and for the thinner grouts. These results contradict previous research findings [43,61], which concluded that, generally, the volume loss of the modified system is reduced after the addition of polymers. Obviously, the addition of high dosages of polymer (especially ER) and PCE superplasticiser increases the dispersion ability, so the cement mixture entraps less water and, consequently, results in the reduction of the final volume [63]. It should be pointed out that the addition of hardener in EMGs did not seem to affect their bleeding.

Figures 9 and 10 depict the setting time of the various examined grouts. Apparently, a retardation of the final setting was caused by the presence of ER or AR, in all grouts [64].

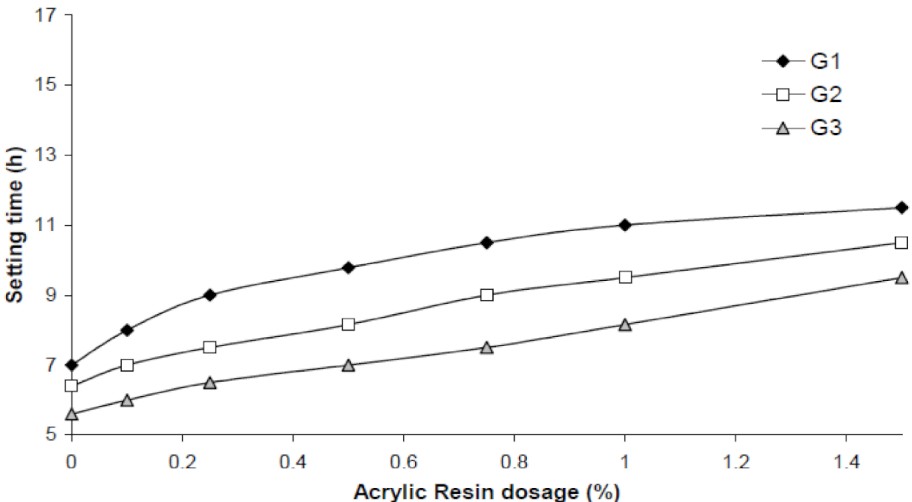

**Figure 9.** Setting time of the unmodified grouts and AMGs.

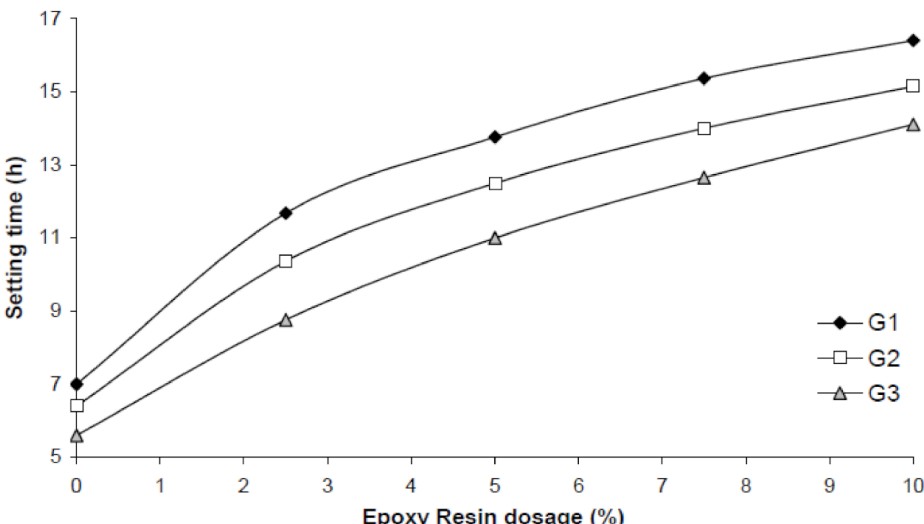

**Figure 10.** Setting time of the unmodified grouts and EMGs.

The setting time increased as the quantity of polymer increased. The extension of setting time was more pronounced in the case of ER and for the thinner grouts. However, despite the retardation effect of both polymers, the setting times did not exceed 24 h, which in construction scenarios represents the maximum setting time for grouting [65]. It should be mentioned that the experimental results indicated that the presence of hardener in EMGs did not induce any additional delay of their final setting.

In order to correlate the development of the strength properties (compressive strength, indirect tensile strength and elastic modulus) of polymer-modified cement grouts made with various dosages of AR or ER (containing hardener) to the descriptor variables (polymer dosage, w/c ratio and curing time), a nonlinear regression analysis was conducted by using the SPSS 17.0 statistical program.

Concerning the compressive strength, indirect tensile strength and elastic modulus of AMGs, the model that gives the best correlation is the following:

$$y = a(b + WC^c)(d + AR^e)(f + AGE^g) + h \qquad (5)$$

where y is the dependent variable corresponding to the examined strength property; a, b, c, d, e, f, g and h are coefficients calculated from the regression analysis; WC is the water to cement ratio; AR is the percentage of the acrylic resin dosage and AGE is the curing time

in days. The values of the regression coefficients and $R^2$, for each mechanical property, are given in Table 11.

**Table 11.** The values of the regression coefficients and $R^2$ for each mechanical property of AMGs.

| Parameter | a | b | c | d | e | f | g | h | $R^2$ |
|---|---|---|---|---|---|---|---|---|---|
| Compressive strength (MPa) | 3379 | −1 | −0.003 | 3.79 | 0.26 | 0.38 | 0.21 | −5.83 | 0.93 |
| Indirect tensile strength (MPa) | −3684 | −0.61 | 1.14 | 2.05 | 0.25 | −1 | $5.6 \times 10^{-4}$ | 1.86 | 0.95 |
| Elastic modulus (GPa) | 156.53 | −0.99 | −0.07 | 4.54 | 0.42 | 0.3 | 0.12 | −1.88 | 0.95 |

The produced models were based on a number of data equal to 252 for each mechanical property.

In the case of EMGs, the model relating the aforementioned strength parameters to the descriptor variables is as follows:

$$y = a(b + cWC^d)(e + AR^f)(g + AGE^h) \qquad (6)$$

where ER is the percentage of the epoxy resin dosage. The values of the regression coefficients and $R^2$, for each mechanical property, are given in Table 12.

**Table 12.** The values of the regression coefficients and $R^2$ for each mechanical property of EMGs.

| Parameter | a | b | c | d | e | f | g | h | $R^2$ |
|---|---|---|---|---|---|---|---|---|---|
| Compressive strength (MPa) | 26.87 | 0.49 | −91.12 | 8.54 | −0.68 | 0.01 | 13.66 | 0.62 | 0.96 |
| Splitting tensile strength (MPa) | 13.6 | 0.99 | −46.14 | 6.7 | −0.96 | 0.004 | 5.88 | 0.49 | 0.92 |
| Elastic modulus (GPa) | −8.95 | 7.35 | −7.34 | −0.01 | 54.62 | 0.64 | −0.78 | 0.03 | 0.92 |

The produced models were based on a number of data equal to 240 for each mechanical property.

The above models satisfactorily reproduce the overall pattern, as is observed from the $R^2$ values, which are very close to unity.

## 5. Conclusions

By considering the information and findings obtained from this thorough experimental research, the following conclusions can be stated:

- All grouts incorporating AR exhibited slightly lower or higher viscosity values than the unmodified grouts. On the contrary, the addition of ER substantially increased the viscosity of all w/c ratios. The higher the ER dosage, the higher the viscosity of the grouts. The addition of hardener did not affect their rheological behaviour.
- Both polymers increased the setting time. However, their retardation effect did not result in prolonged setting times and reduced early strength, except in a few cases when ER was added.
- The strength of all the polymer-modified grouts apparently tends to increase over time, leading to essentially higher values of strength compared to that of the unmodified grouts, a fact suggesting that these materials have great potential for application in the production of high-performance cement grouts.
- ER can be used without a hardener to improve the strength of grouts.

- Both polymers substantially increased the bleeding, which appeared to be more pronounced in the case of ER.
- The accuracy of the regression models, which relates the development of the strength parameters to the descriptor variables, appeared to be satisfactory.

**Author Contributions:** All the co-authors contributed equally to designing the experiments, analysing the data, writing the manuscript, revisions and editing. All authors have read and agreed to the published version of the manuscript.

**Funding:** This research received no external funding.

**Institutional Review Board Statement:** Not applicable.

**Informed Consent Statement:** Not applicable.

**Data Availability Statement:** The data presented in this study are available on request from the corresponding author.

**Conflicts of Interest:** The authors acknowledged no conflict of interest.

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
