# Peer review of "Study on High Performance Polymer-Modified Cement Grouts"

_2673-4109, doi:10.3390/civileng2010008_

Round 1

Reviewer 1 Report

I suppose, the authors want to point out the benefits from using polymer-modified grouts. If there were no benefits, nobody would like to discuss the addition of polymer emulsions to cementitious materials, as they are more expensive, more complicate in handling and more sensitive during application and the paper was of no interest for publication.

According to the findings of this paper, the effects of polymer-modification influence the following items:

  • Viscosity:                 approximately equal or worse (higher shear stresses, AR better than ER). Please add some information according to temperature effects, which might be evident for viscosity (if possible).
  • Setting time:                 worse, but not too much. Again: temperature effects might be evident. Please add information, if possible.
  • Bleeding                 worse
  • Early age strength: approximately equal, slightly worse
  • Final strength: very much better (both, compressive and tensile)

Therefore, the main interest for the use of polymer modification seems to be the improvement in strength. The strength of the unmodified grout significantly depends on the kind of curing, which was not mentioned (or I couldn’t find it) in the paper. Please add detailed information on that.

I suppose, the authors used dry curing, which is very unfortunate for cementitious grout. So the specimen might be impaired by drying shrinkage or internal stresses. I would prefer wet curing for those experiments or at least a very good sealed curing. Without any information on these curing circumstances, the information of the paper might lead to misunderstandings. For me, the most effective benefit of polymer-addition is in its curing effect and shrinkage reduction. However, the paper does not give information on these topics. Please add detailed information, if possible.

Without additional information the paper is insufficient.

Please use a more uniform scaling in the diagrams, especially the same scaling for shear strain rate, bleeding, setting time, ... to better be able to compare between the related diagrams.

Reviewer 2 Report

The paper presents an interesting "Study on high performance polymer-modified cement grouts"

The study is well described in scientific terms and is supported by laboratory tests.

Need more references to this paper, Please try to add some references related to your topic.

Suitable references to this paper:

  1. Fundamental properties of epoxy resin-modified cement https://doi.org/10.1016/j.conbuildmat.2016.08.050
  2. Feasibility Study on the Utilization as Repair Grouting of High Flowable Polymer-Modified Cement Mortar, Adding High Volume Polyacrylic Ester (PAE)    https://doi.org/10.3130/jaabe.7.363
  3. Evaluation of cementitious repair mortars modified with polymers https://doi.org/10.1177/1687814016688584
  4. Concrete repair using two-stage concrete method  DOI: 10.15199/33.2015.08.17
  5. Effect of Silica Fume on two-stage Concrete Strength   IOP Conference Series: Materials Science and Engineering

Reviewer 3 Report

Comments on the Manuscript CIVILENG-1034719

The effect of some admixtures on rheological, setting time, bleeding and strength evolution of cement grouts has been considered in this study. This experimental work accomplishes with standards demanded for a rigorous scientific research. Then, after minor corrections, I could recommend it to be published in the journal CivilEng.

Lines 131-132. How was the device for the rheological measurements calibrated? This information should be incorporated to the Part 3.

Lines 143-147. Equation 1 is directly obtained considering the balance of forces acting on a control volume of the fluid in the flow field (why is the shear stress at the wall of the tube used?). After taking Hagen-Poiseuille equation for the calculation of the viscosity (assumed Newtonian, by the way) the expression for the shear rate (Equation 2) is deduced. Authors should indicate clearly this procedure.

Lines 148-149. Here k and n citation should not appear.

Figure 3a. The behavior of G1 sample without resin is strange. Can the authors explain it?

Lines 161-162. This is Power law equation. As you know, the addition of the yield stress defines Herschel-Bulkley equation. Please, re-write this paragraph to avoid confusion. You say that the yield stress was “very low or negligible” but Figures 2 and 3 show that was just zero. I suppose that you took (0,0) point as an experimental point, but it is not really a measurement. In my opinion, you should eliminate (0,0) “experimental” data from figures.

Conclusion part. Authors should indicate clearly what are conclusions and what are results.

Figures 4-6 appear at the end of the manuscript instead of in the body.

Round 2

Reviewer 1 Report

Thank you for giving additional information. For further investigations it would be of specific interest, to find out the reasons, because of which mechnism the polymer addition is effecting changes in the properties of the grout.